# Effectiveness and Safety of Glucosamine in Osteoarthritis: A Systematic Review

**DOI:** 10.3390/pharmacy11040117

**Published:** 2023-07-14

**Authors:** Nam Xuan Vo, Ngan Nguyen Hoang Le, Trinh Dang Phuong Chu, Huong Lai Pham, Khang Xuan An Dinh, Uyen Thi Thuc Che, Thanh Thi Thanh Ngo, Tien Thuy Bui

**Affiliations:** 1Faculty of Pharmacy, Ton Duc Thang University, Ho Chi Minh City 700000, Vietnam; hoangngan25080177@gmail.com (N.N.H.L.); trinhchu140901@gmail.com (T.D.P.C.); laihuong49@gmail.com (H.L.P.); dinhxuanankhang94@gmail.com (K.X.A.D.); thucuyenpct@gmail.com (U.T.T.C.); thahthah0906@gmail.com (T.T.T.N.); 2Faculty of Pharmacy, Le Van Thinh Hospital, Ho Chi Minh City 700000, Vietnam; thuytienbui2404@gmail.com

**Keywords:** osteoarthritis, effectiveness, safety, glucosamine, systematic review

## Abstract

Knee osteoarthritis is the most popular type of osteoarthritis that causes extreme pain in the elderly. Currently, there is no cure for osteoarthritis. To lessen clinical symptoms, glucosamine was suggested. The primary goal of our systematic review study is to evaluate the effectiveness and safety of glucosamine based on recent studies. Electronic databases such as PubMed, Scopus, and Cochrane were used to assess the randomized controlled trial (RCT). From the beginning through March 2023, the papers were checked, and if they fulfilled the inclusion criteria, they were then examined. The Western Ontario and McMaster Universities Osteoarthritis (WOMAC) and Visual Analog Scale (VAS) scales were considered the main outcome measures. A total of 15 studies were selected. Global pain was significantly decreased in comparison to placebo, as measured by the VAS index, with an overall effect size of standardized mean difference (SMD) of −7.41 ([95% CI] 14.31, 0.51). The WOMAC scale confirmed that pain, stiffness, and physical function had improved, however the effects were insufficient. A statistical update also revealed that there were no reports of serious medication interactions or significant adverse events. To summarize, glucosamine is more effective than a placebo at reducing pain in knee osteoarthritis patients. In long-term treatment, oral glucosamine sulfate 1500 mg/day is believed to be well tolerated.

## 1. Introduction

Around 58 million adults today have osteoarthritis (OA), a degenerative inflammatory disorder of the joint cartilage. By 2040, that number is expected to rise to 78.4 million [1]. Articular cartilage degeneration, subchondral bone remodeling, and synovial low-grade inflammation are anatomical features of OA [2]. This inflammation may result in discomfort, stiffness, and a decreased range of motion, which are finally referred to as arthritic joints [1]. Knee osteoarthritis is the most common type of osteoarthritis that affects the lower limb [3]. That is the location where impairment occurs the most frequently [3]. Mobility declines as a result of the usual symptoms of knee OA, including pain, joint contracture, misalignment, and muscle weakening [4]. These symptoms may eventually raise your chance of being overweight, developing diabetes, and experiencing fractures [1]. Moreover, OA also affects younger people, proving that it is not just a disease of the elderly [2].

The use of both conservative therapy and surgical techniques has been made in the medical treatment of OA [5]. Lessening discomfort, enhancing function and quality of life, and reducing disability are the objectives of treating OA [5]. However, there are presently no disease-modifying therapies available for OA due to inadequate knowledge about pathology. Also, the lack of a biomarker that is sensitive enough to enable detection in the early stages of the disease causes a delay in therapeutic management [2].

The American College of Rheumatology (ACR) and the Osteoarthritis Research Society International (OARSI) recommend patient education and self-management, land-based activity, and dietary weight management for those who are overweight or obese as the main therapy for knee OA [6]. Analgesia, which includes the use of paracetamol, topical and oral nonsteroidal anti-inflammatory medicines (NSAIDs), and opioid medications, and hyaluronic acid (HA) infiltration continue to be the cornerstone of pharmaceutical treatment for symptomatic OA [7,8]. Nevertheless, they are accused of escalating the likelihood of unfavorable occurrences in the gastrointestinal or cardiovascular systems as they solely concentrate on symptomatic relief rather than curing the sickness [9].

This is why symptomatic slow-acting drugs (SYSADOAs), which can alleviate the clinical symptoms of OA with superior tolerance and safety profiles, have recently made waves [9,10]. One of these is glucosamine, a natural substance that ranks among the body’s most common monosaccharides [5]. For more than 50 years, it has been used as a medication to treat OA [2]. Most scientific societies in Europe, but not those in the United States, suggest glucosamine for the treatment of knee problems [2]. Glucosamine sulfate (GS) is one of two medications regarded as a first-line pharmacological treatment for slow-onset, medium- to long-term control of symptoms, following The European Society for Clinical and Economic Aspects of Osteoporosis and Osteoarthritis (ESCEO) advice for the prolonged use of SYSADOAs [11].

In Vietnam, Glucosamine is only indicated as a supporting role in the osteoarthritis therapy process for symptom reduction. Other locations than knee joint are not recommended to utilize glucosamine [12]. In addition, glucosamine is one of the two drugs listed in the health insurance coverage for mild-to-moderate osteoarthritis treatment [13].

Most recent findings from clinical research still demonstrate the controversial debate about glucosamine’s therapeutic effects. To expand the data supporting the safety and effectiveness of glucosamine in the treatment of osteoarthritis, we will conduct a systematic review of recent RCTs in this study.

## 2. Materials and Methods

### 2.1. Search Strategy

This study was designed according to PRISMA guidelines. To conduct our search, we utilized online scientific databases, including PubMed, Cochrane, and Scopus. We limited the search to articles from inception to March 2023. The following keywords were combined to retrieve the studies: “glucosamine”, “osteoarthritis”, “knee osteoarthritis”, “effectiveness”, “effect”, “safety”, “placebo”, “randomized”, “double-blind”, and “RCT” using a Boolean connector. Additionally, further citations were discovered after screening the reference lists of all the articles obtained. The only participants in our search are humans, and we exclusively use English-language articles.

### 2.2. Selection Criteria

Studies were included if they met the following criteria: (1) the research was a randomized placebo-controlled trial with either a parallel or cross-over design, either for effectiveness or safety; (2) patients had a diagnosis of knee, hip, or hand osteoarthritis at minimum; (3) at least indicates a comparison between oral glucosamine with chondroitin and placebo. (4) Sufficient data about pain, physical function, and stiffness based on the WOMAC index or VAS scale at the end of the treatment.

The other studies were excluded by: (1) studies of non-randomized and/or uncontrolled trials; (2) comparing glucosamine in combination form to other drugs; (3) a lack of a placebo control group for glucosamine; and (4) Unclear information about VAS or WOMAC index sub-scores at the end of treatment.

### 2.3. Data Analysis and Outcome Measure

In each article, the following information was extracted: (1) the first author’s name; (2) the publication year; (3) the design of study; (4) the type of OA; (5) number of participants in the intervention and control groups; (6) the duration of treatment; (7) the type of outcome measure (VAS/WOMAC); and (8) the improvement of treatment or not.

The Western Ontario and McMaster Universities Osteoarthritis (WOMAC) pain scale was widely used as a self-assessment tool for pain, physical function pain, and stiffness pain in patients with osteoarthritis, especially the knee and hip [14]. The Visual Analog Scale (VAS) is a scale to show the level of pain from 0 to 100, which represents a range between “no pain” and “worst pain” [15].

The primary outcomes were the degree of pain (based on WOMAC/VAS or other tools), the improvement in function, and the stiffness score from baseline to the end of treatment. Safety was considered a secondary outcome. SMD divides the pooled SD of the differences between two interventions to represent the extent of the intervention impact in each study compared to the variability observed in that study [9]. In order to pool the data, outcomes measured on different scales were standardized. Standardized mean difference (SMD) estimations and 95% confidence intervals (95% CI) were used to present the results for the comparative effect between the glucosamine group and the control group.

In this study, the authors extracted data and tried to summarize the results based on the categories from the selected studies. In these studies, the researchers used the effect size transformed back to the different units of the WOMAC or Visual Analogue Scale (VAS), the most commonly used scale based on a media pooled standard deviation to assess pain. The heterogeneity and the clinically substantial improvement were considered criterion to clarify the improvement of treatment. Moreover, the number of adverse events and their frequency were counted to show the safety of glucosamine.

### 2.4. Risk of Bias

To assess the methodological quality of the included studies, the Cochrane Risk of Bias Tool was utilized [9]. Based on the criteria, the selected studies were scored as ‘yes’ (low risk of bias), ‘no’ (high risk of bias), or ‘unclear’ [16]. A study with a low risk of bias was defined as fulfilling six or more of the criteria items [16].

## 3. Results

### 3.1. Selection of Study Process

Combining these keywords “glucosamine”, “osteoarthritis”, “effect”, “effectiveness” and “safety”, a total of 797 articles were identified through PubMed, Cochrane, and Scopus databases, as shown in Figure 1. We then eliminated duplicates and examined the titles and abstracts of all the studies by 4 investigators in 2 independent working groups. Any papers that demonstrated a lack of relevance to our topic were disqualified. When any disagreement occurs, the researchers discuss and agree on the results with 2 other investigators based on the selection criteria. As a result, 797 articles were screened by title and abstract, and only 53 publications were eligible to be reviewed. Through the evaluating process, those articles not fulfilling the criteria included unsuitable contents (721 articles), non-English writing (8 articles), and full-text unavailability (03 articles). Additionally, we came across 1 article finding from references [17]. Overall, 15 publications were deemed sufficient for data extraction in Table 1.

The study’s data were compiled from 15 randomized, placebo-controlled articles. In which almost all articles use the double-blind method. The study included 2859 subjects who completed it. Of these, 1428 were in the control group, and 1431 were in the Glucosamine group. Clinical studies were published between 2000 and 2015. The studies were followed up for 1.5 to 36 months. The dose of glucosamine used in these clinical studies is 1500 mg/day. In the studies we have synthesized, we have used two forms of glucosamine: glucosamine sulfate and glucosamine hydrochloride [19], or both forms [26]. These studies were focused mainly on the knee and used the WOMAC scale as the primary outcome measure.

### 3.2. Risk of Bias Assessment

Evaluation on Methodological quality was summarized in Appendix A. A low risk of bias for selective outcome reporting was confirmed in most of the included studies, aside from Nieman et al.’s 2013 trial [17].

### 3.3. Effectiveness of Glucosamine on Knee Osteoarthritis

According to the VAS scale (Table 2), Glucosamine showed improvement versus placebo, with an overall difference −7.41 ([95% CI] −14.31, −0.51), judging by the global pain.

The WOMAC scale is divided into three subscales: pain, physical function, and stiffness. The results are shown in Table 3. On the knee joint, all three categories showed favorable statistical change but did not contribute considerable benefit, as the effect size indicated −0.04 ([95% CI]–0.13, 0.06) for pain (Appendix A), −0.07 ([95% CI]–0.17, 0.03) for physical function (Appendix A), and −0.30, ([95% CI] 0.82, 0.21) for stiffness (Appendix A), respectively. It can also be noted that Glucosamine did not affect the total WOMAC score, with a SMD −2.27 ([95% CI]–5.21, 0.66).

### 3.4. Safety

#### 3.4.1. Adverse Events

Appendix A displays the safety and tolerability outcomes, including patients withdrawn because of adverse events. There was no discernible difference between glucosamine and placebo in terms of the frequency of adverse events. In which the incidence of adverse events associated with using glucosamine was slightly lower than that of using placebo.

The rate of some common adverse events is represented in Appendix A. Most adverse events associated with glucosamine were located in the gastrointestinal system. However, there were no serious adverse events.

#### 3.4.2. Drug Interactions

Most studies showed no serious drug interactions exist [32,33]. However, there were some studies stating that using glucosamine with warfarin may increase the anticoagulation effect, but still, further research is required for more information [34,35,36]. On the other hand, there was also a theory that glucosamine reduced the effectiveness of diabetes medications [33,35].

## 4. Discussion

Our search tends to focus on studies using those two indexes because they are the most frequently used to assess the effects of glucosamine. Notably, measuring pain intensity is a commonality between WOMAC and VAS. Nevertheless, based on our research, their results showed a sizable gap. At the end of the follow-up, the VAS score had significantly improved, but the positive change in WOMAC pain was too slight to be influential. Additionally, neither the overall WOMAC score nor any of its subscales established a discernible clinical improvement. It is important to remember that the VAS global pain subscale was thought to have better assay sensitivity than the WOMAC pain subscale, which exhibited valuable authenticity as a primary result measure [37]. Even though glucosamine has been found to mitigate discomfort, the scores in the placebo group also improved. This improvement could be the result of the arthritis’ normal progression or the effects of a placebo. Other musculoskeletal structures have been affected by this issue as well.

Most of the articles we collected utilized Glucosamine sulfate rather than Glucosamine hydrochloride. Despite inconsistent findings across studies, there was a tendency that glucosamine sulfate portrayed a noticeable and clinically meaningful impact on OA at high doses [38]. According to the hypothesis, glucosamine works by modifying the O-GlcNAcylation pathway, a reversible post-translational modification similar to phosphorylation that regulates protein activity, location, or stability depending on the availability of glucose. This is performed by attaching N-Acetylglucosamine to the serine or threonine residues of cytosolic or nuclear proteins [38].

Our research showed that glucosamine was more effective than placebo at lessening pain. This differed from earlier meta-analyses that yielded conflicting findings about the symptomatic effectiveness of glucosamine in treating knee osteoarthritis [9,16]. It should be highlighted that the glucosamine-modifying therapeutic effects were centered mostly in the short-term RCTs, since the long-term ones found no superior advantage versus placebo [18,39,40]. Given that osteoarthritis is a chronic illness, glucosamine should be evaluated over a longer period of time to ascertain how it affects the human body. On the other hand, one of the key reasons that influences one’s decision to look for medications is pain degree [41]. As a result, our study results could perhaps help individuals with knee osteoarthritis enhance their quality of life.

Using oral glucosamine sulfate (1500 mg/day) has no significant difference in the frequency of adverse events compared to placebo. In addition, adverse events reported in the glucosamine group were slightly lower than those reported in the placebo group, with RR = 0.99 (95% CI 0.66, 1.23), as shown in Appendix A. According to Appendix A, most adverse events affected the gastrointestinal tract (including abdominal pain, diarrhea, and dyspepsia). There were no serious adverse events reported over a period of 3–4 months based on the study duration of the majority of studies. On the other hand, in another trial lasting 2–3 years of using Glucosamine, there were also no serious adverse events occurring [42]. Some people withdrew from the study due to serious adverse events that were not related to the study treatment, mostly because of pre-existing or concurrent diseases.

Glucosamine has no serious drug interactions. However, using warfarin and glucosamine together may boost the anticoagulant impact, according to several studies, although additional research is required. A different hypothesis suggested that glucosamine decreased the efficiency of diabetes medicines. If patients are currently using diabetes drugs or warfarin, they should inform the doctor in case drug interactions occur.

The majority of our articles are limited to 6 months of treatment (12/15 articles), and they were conducted in 2015 or earlier. Our study mainly focused on RCT studies, so no new studies have been conducted since 2015 because of the effectiveness and safety of glucosamine use and the fact that there are almost no serious side effects or adverse events. The observation of joint space narrowing was not mentioned. We focus on the symptomatic effect, assessed by VAS and WOMAC indexes, rather than the structural effect of glucosamine. In addition, restrictions on time might contribute to incomplete data on the long-term effectiveness and safety of glucosamine. Overall, most research has small sample sizes, which could limit the ability to comprehend the outcomes that have been displayed [41]. Ultimately, potential sources of inter-study heterogeneity that could have impacted our findings included the severity of OA, different formulations, and the duration of therapies.

## 5. Conclusions

Glucosamine showed a decrease in global pain based on VAS scores in knee osteoarthritis patients. Moreover, glucosamine is safe. It does not cause serious side effects and has no serious drug interactions. However, further studies of glucosamine are needed in patients who are taking warfarin or diabetic medicines. At the same time, doctors should also inform patients about possible interactions when using these drugs together.

## 6. Future Directions

The effects of glucosamine use have been shown to be effective in the short-term. However, since osteoarthritis is a chronic disease, patients with osteoarthritis will take medication for most of their lives if no alternative therapy is available. Therefore, future research should focus on performing long-term studies on the effects, effectiveness, and safety of glucosamine to assess the effectiveness of this use more accurately.

## Figures and Tables

**Figure 1 pharmacy-11-00117-f001:**
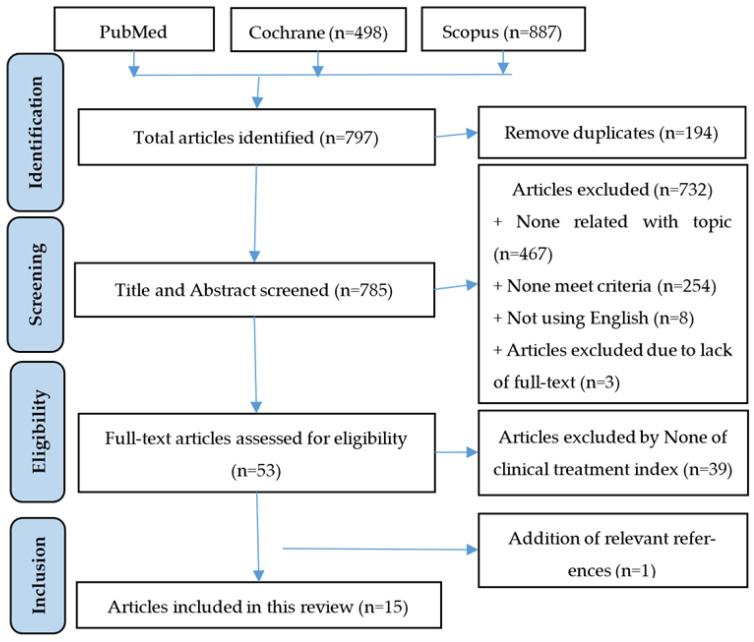
Flow diagram of the study selection process.

**Table 1 pharmacy-11-00117-t001:** Characteristics of the included studies.

Author, Year, and Country	Study Design	OA Type	Intervention	N in Control Group	N in Glucosamine Group	Follow-Up (Months)	* Age	Tool to Evaluate Degree of Pain	Primary Outcome
Fransen et al., 2015, Australia [18]	Randomized, placebo-controlled, double-blind trial	Knee	GS vs. GS + CS vs. CS vs. P	151	152	24	61.2 ± 7.7 60.6 ± 8.1	WOMAC	Improvement
Kwoh et al., 2014, USA [19]	Randomized, placebo-controlled, double-blind trial	Knee	GS + P	103	98	6	52.17 ± 6.05 52.29 ± 6.72	WOMAC	Improvement
Madhu et al., 2013, India [20]	Randomized, placebo-controlled, single-blind trial	Knee	GS vs. P vs NR-INF-02 vs. NR-INF-02 + GS	30	30	1.5 (42 days)	56.80 ± 7.99 56.77 ± 9.98	WOMAC, VAS, CGIC	Improvement
Nieman et al., 2013, USA [17]	Randomized, placebo-controlled, double-blind trial	Knee, Hip Ankles Shoulders Hand	GS vs. P	51	101	2	57.6 ± 0.9 58.3 ± 0.8	WOMAC, VAS, SF-36, 6-MWD	Improvement
Petersen et al., 2011, Denmark [21]	Randomized, placebo-controlled, double-blind trial	Knee	GS vs ibuprofen vs. P	12	12	3	62.2 ± 3.4 63.1 ± 4.7	VAS	Improvement
Giordano et al., 2009, Italia [22]	Randomized, placebo-controlled, double-blind trial	Knee	GS vs. P	30	30	3	57.2 ± 7.2 58.0 ± 8.3	WOMAC, VAS	Improvement
Frestedt et al., 2008, USA [23]	Randomized, placebo-controlled, double-blind trial	Knee	GS vs. P vs Aquamin vs. GS + aquamin	16	19	3	59.2 ± 8.3 58.9 ± 7.4	WOMAC, 6-MWD	Improvement
Herrero-Beaumont et al., 2007, Spain [24]	Randomized, placebo-controlled, double-blind trial	Knee	GS vs. P	104	106	6	63.4 ± 6.9 64.5 ± 7.2	WOMAC	Improvement
Clegg et al., 2006, USA [25]	Randomized, placebo-controlled, double-blind trial	Knee	GH vs. CS vs GH + CS vs. P vs Celecoxib	313	317	6	58.6 ± 10.2 58.2 ± 9.8	WOMAC	Improvement
Cibere et al., 2004, Canada [26]	Randomized, placebo-controlled, double-blind trial	Knee	GS vs. P	66	71	6	64 (40–83) ^a^ 65 (43–88) ^a^	WOMAC, EQ-5D	Improvement
McAlindon et al., 2004, USA [27]	Randomized, placebo-controlled, double-blind trial	Knee	GH vs. P	104	101	3	ND ND	WOMAC	Improvement
Hughes et al., 2002, UK [28]	Randomized, placebo-controlled, double-blind trial	Knee	GS vs. P	40	40	6	** 62.28 ± 9.12	WOMAC, VAS, McGill pain questionnaire	Improvement
Pavelka et al., 2002, Czech Republic [29]	Randomized, placebo-controlled, double-blind trial	Knee	GS vs. P	101	101	36	61.2 ± 7.2 63.5 ± 6.9	WOMAC	Improvement
Reginster et al., 2001, Belgium [30]	Randomized, placebo-controlled, double-blind trial	Knee	GS vs. P	106	106	36	66.0 ± 8.1 65.5 ± 7.5	WOMAC	Improvement
Rindone et al., 2000, USA [31]	Randomized, placebo-controlled, double-blind trial	Knee	GS vs. P	49	49	2	63 ± 12 64 ± 11	VAS	Improvement

Abbreviations: ND: no data; GS: glucosamine sulfate; GH: glucosamine hydrochloride; CS: chondroitin sulfate; P: placebo; * Age: the upper number indicates Glucosamine group; the lower number indicates Placebo group; ** representative for both glucosamine and placebo group; ^a^ median (IQR); IQR: interquartile range.

**Table 2 pharmacy-11-00117-t002:** The mean difference and 95% CI for the effect of placebo versus glucosamine on VAS.

Study and Year	Placebo			Glucosamine			Std. Mean Difference	
	Total	Mean	SD	Total	Mean	SD	Weight	IV, Random, 95% CI
Fransen et al., 2015 [18]	151	−7.2	33.8	152	−8.6	24.5	4.1%	−1.40
								[−8.05, 5.25]
Madhu et al., 2013 [20]	29	−15.5	18.3	24	−31.7	19.0	3.5%	−16.20
								[−26.31, −6.09]
Petersen et al., 2011 [21]	12	−1.9	10.7	12	−16.8	17.3	3.3%	−14.90
								[−26.41, −3.39]
Giodarno et al., 2009 [22]	30	0.3	10.8	30	−16.6	22.4	3.7%	−16.90
								[−25.80, −8.00]
Clegg et al., 2006 [25]	313	−16.6	25.2	317	−16.0	26.9	4.5%	0.60
								[−3.47, 4.67]
Rindone et al., 2000 [31]	49	−15.0	23.4	49	−15.0	26.6	3.6%	0.00
								[−9.92, 9.92]

**Table 3 pharmacy-11-00117-t003:** The mean difference and 95% CI for the effect of placebo versus glucosamine on WOMAC.

Study and Year	Placebo			Glucosamine			Std. Mean Difference	
	Total	Mean	SD	Total	Mean	SD	Weight	IV, Random, 95% CI
Kwoh et al., 2014 [19]	103	−19.1	20.1	98	−15.1	19.3	9.9%	4.00
								[−1.45, 9.45]
Madhu et al., 2013 [20]	29	−9.3	11.4	24	−23.4	17.1	7.3%	−14.10
								[−22.0, −6.10]
Frestedt et al., 2008 [23]	9	−5.9	16.9	14	−10.5	15	3.8%	−4.60
								[−18.15, 8.95]
Herrero-Beaumont et al., 2007 [24]	70	−11.7	14.3	78	−17.3	13.3	11.0%	−5.60
								[−10.06, −1.14]
Cibere et al., 2004 [26]	66	3.4	18.1	71	3.2	15.5	9.7%	−0.20
								[−5.86, 5.46]
McAlindon et al., 2004 [27]	104	7.8	13.5	101	7.8	13.1	11.9%	0.00
								[−3.64, 3.64]
Pavelka et al., 2002 [29]	55	−4.7	5.9	66	−7.7	7.1	13.2%	−3.00
								[−5.32, −0.68]
Regisnter et al., 2001 [30]	71	−0.6	19.6	68	−0.2	19.2	8.8%	0.40
								[−6.05, 6.85]

## Data Availability

Data was collected in PubMed, Scopus and Cochrance database.

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
