# Peer review of "Effectiveness and Safety of Glucosamine in Osteoarthritis: A Systematic Review"

_pharmacy, 2023, doi:10.3390/pharmacy11040117_

Round 1

Reviewer 1 Report

The authors should spell out the keywords in abstract

The structure of the articles does not like to review articles. The authors should check the authors' guidelines.

Figure 1, the font size is very small to read please make sure.

How the statistical analysis has benefited in tables there is no information about the analysis.

In table title some of references have cited is there normal?

The discussion part should be added the future directions.

Reviewer 2 Report

Very interesting research and applicable for the pharmacy practice. 

The section Materials and Methods: please provide more information how many investigators were included in the study and describe it.

The section Results is quite messy, I suggest to change tables for forests plots. This will make the results more visible and understandable. Moreover, Table 1 is missing some data. I suggest to describe more in detail the measured outcome (primary and/or secondary) and add results and overall assessment of results (positive or negative). This Table 1 can then become Appendix one if the authors conclude that it will be to big to be included in the summary. 

Throughout the text there are small mistakes and words missing, please correct.

Reviewer 3 Report

In the review: “Effectiveness and safety of Glucosamine in osteoarthritis: A literature review”, the authors discussed about the efficacy and safety of glucosamine of OA management by checking several recent studies.

Overall, this manuscript results very interesting, the authors clearly explain the rational of the study and discussed the topic point by point.

However, we would like to invite the authors  to clarify some minor points:

 1.       Please check the check punctuation and spaces;

2.       Among the introduction, the authors discussed about the different types of OA therapy, they should also in general cite the use of hyaluronic acid infiltration;

3.      Among the introduction the concept of SYSADOAs, in this respect the should deepen the topic  by also briefly  introducing the use of glucosamine. In this context the following reference should be useful; Stellavato A, Restaino OF, Vassallo V, Cassese E, Finamore R, Ruosi C, Schiraldi C. Chondroitin Sulfate in USA Dietary Supplements in Comparison to Pharma Grade Products: Analytical Fingerprint and Potential Anti-Inflammatory Effect on Human Osteoartritic Chondrocytes and Synoviocytes. Pharmaceutics. 2021 May 17;13(5):737. doi: 10.3390/pharmaceutics13050737. PMID: 34067775; PMCID: PMC8156081;

4.    Figure 1; the authors should use a bigger font, it is difficult to read in this form;

5.     Table 1; the authors should try to use a smaller font or to enlarge the columns table;

6.  In conclusion, glucosamine should be superior to other natural molecules in OA management? If yes why, please deepen this concept.

minor errors concerning the spelling are present

Round 2

Reviewer 1 Report

Corrections have been made.

Author Response

Dear Reviewer 1,

Thank you very much for helping us during the review time

Best regards,

Reviewer 2 Report

Response to The author added on information in the first paragraph of part 3.1 (page 4) - I cannot find the added information in the new manuscript version. Please recheck and add the information.

Author Response

Dear Reviewer 2,

Thank you very much for your review time. I had checked and updated follow:

Previous manuscript draft:

Combining these keywords "glucosamine," "osteoarthritis," "effect," and "safety”, a total of 797 articles were identified through PubMed, Cochrane, Scopus databases, as shown on Figure 1. We then eliminated duplicates and examined the titles and abstracts of all the studies. Any papers that demonstrated a lack of relevance to our topic were disqualified. As a result, 53 publications were eligible to be reviewed. Through the evaluating process, those articles not fulfilling criteria included: unsuitable contents, non-English writing, and full-text unavailability. Additionally, we came across 1 article finding from references. Overall, 15 publications were deemed sufficient for data extraction.

Updated manuscript draft:

"Combining these keywords "glucosamine," "osteoarthritis," "effect," and "safety”, a total of 797 articles were identified through PubMed, Cochrane, Scopus databases, as shown on Figure 1. We then eliminated duplicates and examined the titles and abstracts of all the studies by 04 investigators in 02 independent working groups. Any papers that demonstrated a lack of relevance to our topic were disqualified. When any disagreement occurs, the researchers discuss and agree on the results with 02 other investigators based on the selection criteria. As a result, 797 articles were screened by title and abstract. 53 publications were eligible to be reviewed. Through the evaluating process, those articles not fulfilling criteria included: unsuitable contents (721 articles), non-English writing (08 articles), and full-text unavailability (03 articles). Additionally, we came across 01 article finding from references [17]. Overall, 15 publications were deemed sufficient for data extraction in table 1"

Once again, thank you very much. Please feel free to let your comments.

Best regards,